# *R. verniciflua* and *E. ulmoides* Extract (ILF-RE) Protects against Chronic CCl_4_-Induced Liver Damage by Enhancing Antioxidation

**DOI:** 10.3390/nu11020382

**Published:** 2019-02-12

**Authors:** Hwa-Young Lee, Geum-Hwa Lee, Young Yoon, Han-Jung Chae

**Affiliations:** 1Department of Pharmacology and New Drug Development Institute, Chonbuk National University Medical School, Jeonju, Chonbuk 561-180, Korea; youngat84@gmail.com; 2Non-Clinical Evaluation Center, Biomedical Research Institute, Chonbuk National University Hospital, Jeonju, Chonbuk 561-180, Korea; heloin@jbnu.ac.kr; 3Imsil Cheese & Food Research Institute, Doin 2-gil, Seongsu-myeon, Imsil-gun, Chonbuk 55918, Korea; kuburi79@icf.re.kr

**Keywords:** *R. verniciflua*, *E. ulmoides*, carbon tetrachloride, oxidative stress, lipogenesis, lipid peroxidase

## Abstract

This study aimed to characterize the protective effects of *R. verniciflua* extract (ILF-R) and *E. ulmoides* extract (ILF-E), the combination called ILF-RE, against chronic CCl_4_-induced liver oxidative injury in rats, as well as to investigate the mechanism underlying hepatoprotection by ILF-RE against CCl_4_-induced hepatic dysfunction. Chronic hepatic stress was induced via intraperitoneal (IP) administration of a mixture of CCl_4_ (0.2 mL/100 g body weight) and olive oil [1:1(v/v)] twice a week for 4 weeks to rats. ILF-RE was administered orally at 40, 80, and 120 mg/kg to rats for 4 weeks. Alanine transaminase (ALT), aspartate transaminase (AST), gamma-glutamyl transpeptidase (GGT), and lipid peroxidation assays were performed, and total triglyceride, cholesterol, and LDL-cholesterol levels were quantified. Furthermore, ER stress and lipogenesis-related gene expression including sterol regulatory element-binding transcription factor 1 (SREBP-1), fatty acid synthase (FAS), and P-AMPK were assessed. ILF-RE markedly protected against liver damage by inhibiting oxidative stress and increasing antioxidant enzyme activity including glutathione (GSH), glutathione peroxidase (GPx), superoxide dismutase (SOD), and catalase. Furthermore, hepatic dyslipidemia was regulated after ILF-RE administration. Moreover, hepatic lipid accumulation and its associated lipogenic genes, including those encoding SREBP-1 and FAS, were regulated after ILF-RE administration. This was accompanied by regulation of ER stress response signaling, suggesting a mechanism underlying ILF-RE-mediated hepatoprotection against lipid accumulation. The present results indicate that ILF-RE exerts hepatoprotective effects against chronic CCl_4_-induced dysfunction by suppressing hepatic oxidative stress and lipogenesis, suggesting that ILF-RE is a potential preventive/therapeutic natural product in treating hepatoxicity and associated dysfunction.

## 1. Introduction

The liver is a vital metabolic organ in various animal species for biotransformation and detoxification of endogenous and exogenous harmful substances [1,2]. The mechanism underlying oxidative stress-induced liver damage involves imbalance of oxidation and antioxidant systems, thereby generating excessive free radicals and reducing antioxidant capacity [2]. Furthermore, liver diseases associated with oxidative stress facilitate the pathogenesis of hepatic fibrosis, liver cirrhosis, and even hepatocellular carcinoma [2,3]. Carbon tetrachloride (CCl_4_), an industrial solvent, is a potent hepatotoxic agent used extensively in animal models to induce acute and chronic liver injury [4]. CCl_4_ itself does not have cytotoxic effects on the liver; however, its metabolites ·CCl_3_ and ·OOCCl_3_ in hepatic parenchyma cells formed by cytochrome P_450_-dependent monooxygenases cause hepatotoxicity [5]. Notwithstanding extensive advancements in modern pharmacotherapies to treat liver diseases, these drugs are occasionally inadequate and have adverse effects, especially when administered for a long period. Therefore, development of new, more effective, and safer functional foods has important implications in treating liver disease.

Traditional herbal formulations have numerous biological activities with associated health effects. Combinations of various herbal extracts are frequently used to achieve a synergistic effect. The use of herbal combinations has been reported in Chinese [6] and Thai traditional medicine [6]; however, scientific evidence is lacking in regards to their potential bioactive compounds, safety information, and therapeutic benefits. Recent studies have reported the antioxidant properties of polyherbal remedies prescribed as traditional medicine, such as Wuzi Yanzong (Chinese medicine) [7] and Sahatsatara (Thai traditional medicine) [6]. Furthermore, treatment with combined formulations has been frequently applied in human diseases [8,9,10].

*Rhus verniciflua* (RV), commonly known as the lacquer tree, reportedly exhibits various biological activities, including antioxidant, anticancer, antimicrobial, anti-inflammatory, and inhibitory α-glucosidase effects [11,12,13]. These pharmaceutical activities are mediated by the abundant flavonoids and polyphenols in RV, including fustin, fisetin, quercetin, butein, sulfuretin, and ethyl gallate. RV reportedly exerts regulatory effects against altered hepatic metabolism and associated dysfunction [14,15]. Furthermore, the *Eucommia ulmoides* cortex is extensively used to improve liver steatosis and is also considered a functional health food [16,17,18]. *E. ulmoides* reportedly contains polyphenolics, flavonoids, and triterpines as its chemical constituents [19]. Recently, a controlled pilot study has reported the efficacy of an *E. ulmoides*-containing herbal mixture by demonstrating its regulatory effect on alanine aminotransferase (ALT) in non-alcoholic steatohepatitis (NASH) patients [20]. We previously reported that a combined formulation of RV and EU extracts, named ILF-RE herein, displayed synergistic protective effects as opposed to each individual extract (RV or EU) in an in vitro model of CCl_4_-induced hepatotoxicity [21]. However, the efficacy of the combined extract still needs to be determined by an in vivo model of hepatotoxicity along with its detailed underlying mechanism. Based on the established hepatoprotective effects of RV and EU, the present study aims to identify herbal extracts exhibiting enhanced hepatoprotective biological effects when combined with ILF-R and ILF-E. These extract formulations can be potentially used as hepatoprotective agents. 

Herein, we investigated the role of ILF-RE in hepatoprotection in a rat model of CCl_4_-induced liver damage. We hypothesized that ILF-RE protects the liver from CCl_4_-induced damage by reducing oxidative stress, decreasing lipid peroxidation, and suppressing glutathione activation.

## 2. Materials and Methods

### 2.1. Materials

CCl_4_ was purchased from Sigma-Aldrich (St. Louis, MO, USA). Malondialdehyde (MDA) and glutathione (GSH) detection kits were obtained from BioVision (Mountain View, CA, USA). Dihydroethidium (DHE) was obtained from Life Technologies (Carlsbad, CA, USA). Aspartate aminotransferase (AST), alanine aminotransferase (ALT), total cholesterol, triglyceride, HDL cholesterol, and LDL-cholesterol detection kits were obtained from Asan Pharmaceutical Company (Seoul, Korea).

### 2.2. Preparation of ILF-E and ILF-R

*R. verniciflua* and *E. ulmoides* combined extract (ILF-RE) was obtained from the Imsil Cheese & Food Research Institute (Imsil-gun, Jeollabuk-do, Korea) [21]. *R. verniciflua* was finely pulverized, extracted with boiling distilled water, concentrated under reduced pressure, using a rotary evaporator, and lyophilized to obtain dried *R. verniciflua extract* (ILF-R). The yield of the dried *R. verniciflua* extract was 4%. Dried *E. ulmoides* extract was obtained with boiling distilled water, concentrated under reduced pressure using a rotary evaporator, and lyophilized. The yield of the dried *E. ulmoides* extract was 10%. Each extract was mixed at a ratio of 1:1 to prepare the combined ILF-RE.

### 2.3. Analysis of Cell Viability

Primary hepatocytes were cultured at 37 °C in a humidified atmosphere of 5% CO_2_. Culture media were added to adjust the concentration of cancer cells to a logarithmic growth phase of 2 × 10^4^/dish. The cells were seeded in a 96-well culture plate by adding 50 μL per well, and the plate was incubated in an atmosphere of 5% CO_2_ at 37 °C for 24 h. After the supernatant was removed, the blank control group was then mixed with MTT solution and incubated for 4 h. One hundred microliters of DMSO was added to the blank control group after the supernatant was removed, and shocked for 30 min. A spectraMax 190 Microplate Reader (Molecular Devices, Mississauga, ON, Canada) was used as the enzyme standard instrument to detect at 570 nm.

### 2.4. In Vitro Assays for Hepatoprotective Effects of ILF-RE

The hepatoprotective effects of ILF-RE on primary hepatocytes were determined as follows: Normal control cells were incubated with EMEM in DMSO (0.05% v/v) for 6 h. To establish a cellular model of hepatotoxicity, cells were incubated with EMEM in DMSO (0.05% v/v) for 6 h and then treated with EMEM supplemented with 20 mM CCl_4_ for 6 h. For ILF-RE treatment, cells were incubated with EMEM at 25, 50, and 100 μg/mL for 30 min and then treated with 20 mM CCl_4_ for 6 h. 

### 2.5. DPPH Radical Scavenging Assays

Radical scavenging activity of the mulberry extracts was determined as described by Blois [22]. The extract (0.5 mL) and 0.2 mM butylated hydroxyanisole (2 mL) were transferred to separate test tubes, followed by addition of 2 mL of a 0.1-mM ethanol solution of 2,2-diphenyl-1-picrylhydrazyl and vigorous agitation. The tubes were then allowed to stand at 20 °C for 30 min. The control was prepared similarly but without any extract or ethanol. Changes in the absorbance of the prepared samples were determined spectrophotometrically at 517 nm, and radical scavenging activity was estimated as the inhibition percentage, in accordance with the following formula: [(control absorbance-sample absorbance)/(control absorbance)] × 100.

### 2.6. Animal Treatment and Care

Sprague Dawley male rats weighing 250–270 g were obtained from Samtako (Daejeon, Korea) and divided into 6 groups. Rats were maintained on a 12:12 h light, with the dark cycle (lights on at 06:00) in stainless-steel wire-bottomed cages, where they were allowed to acclimate under laboratory conditions for at least 1 week before experiments. Rats were administered an appropriate diet with ad libitum access to water and were weighed weekly. Rats were intraperitoneally (IP) administered with a mixture of CCl_4_ (0.2 mL/100 g body weight) and olive oil [1:1(v/v)] twice a week for 4 weeks. Simultaneously, ILF-RE was orally administered daily at dose of 40, 80, or 120 mg/kg body weight. All animal experiments in this study were performed in accordance with the regulations described in the Care and Use of Laboratory Animals guide of Chonbuk National University Hospital. All procedures were also approved by the Institutional Animal Care and Use Committee of Chonbuk National University Hospital for the animal center (IACUC protocol cuh-IACUC-2018-4).

### 2.7. Sample Collection

Liver and blood samples were harvested from all euthanized animals. Whole blood was immediately placed on ice in a centrifuge tube for 30 min and centrifuged at 7168× *g* for 10 min. Serum was transferred to 1.5-mL tubes and stored at −75 °C. All harvested liver tissue samples were immediately stored at −75 °C.

### 2.8. Histological Analysis

Liver tissue samples were fixed in a 10% formalin solution and embedded in paraffin; after conventional histological tissue preparation, 5-μM-thick sections were cut, adhered to microscopic slides, and stained with hematoxylin and eosin. Thereafter, the extent of CCl_4_-induced damage was estimated on the basis of pathological changes in liver tissue sections under a light photomicroscope. Hepatic steatosis was assessed via Oil Red O staining. Briefly, liver cryo-sections were fixed for 10 min in 60% isopropanol, followed by staining with 0.3% Oil Red O in 60% isopropanol for 30 min, and washing with 60% isopropanol. Sections were counterstained with Gill’s hematoxylin, washed with 4% acetic acid, and mounted in an aqueous solution. Stained sections were quantified via histomorphometric analysis.

### 2.9. Blood Biochemical Marker Assays

Serum ALT, AST, and gamma-glutamyl transpeptidase (GGT) activity was determined via a colorimetric procedure using commercially available detection kits. Glutathione peroxidase (GPx), superoxide dismutase (SOD), and catalase (CAT) activities and malondialdehyde (MDA) levels were measured using commercial assay kits in accordance with the manufacturer’s instructions (BioVision, Inc., Mountain View, CA, USA).

### 2.10. Determination of Total Lipid, Triglyceride, and Cholesterol Levels

To determine total lipid levels, liver homogenates were extracted in accordance with the modified Bligh and Dyer procedure [23]. Briefly, samples were homogenized with chloroform–methanol–water (8:4:3), shaken at 37 °C for 1 h, and centrifuged at 1100× *g* for 10 min. The bottom layer was harvested for hepatic lipid analysis. Triglyceride, total cholesterol, and cholesterol content were determined using kits from Asan Pharmaceutical Company (Seoul, Korea) in accordance with the manufacturer’s instructions.

### 2.11. Analyses of Lipid Peroxidation

Lipid peroxidation was assessed in liver tissue, using a lipid hydroperoxide assay kit purchased from Cayman Chemicals (Ann Arbor, MI, USA) in accordance with the manufacturer’s instructions. In this assay, lipid hydroperoxide was extracted from the samples in chloroform, using the extraction buffer from the kit. The chromogenic reaction was carried out at 37 °C for 5 min, and the absorbance of each well was determined at 500 nm, using a 96-well plate spectrometer (SpectraMax 190, Molecular Devices Corp., Sunnyvale, CA, USA). The standard used was 13-Hydroperoxy-octadecadienoic acid. Cellular lipid hydroperoxide levels were determined as described by the manufacturer.

### 2.12. Immunoblotting

Liver tissue homogenates were prepared in RIPA buffer (50 mM Tris, pH 8.0; 150 mM NaCl; 2 mM EDTA; 1% Nonidet P-40; 0.1% SDS) supplemented with a Protease Inhibitor Cocktail Tablet (Roche, Indianapolis, IN, USA) and Phosphatase Inhibitory Cocktails 2 and 3 (Sigma-Aldrich). Lysates were cleared via centrifugation and analyzed via polyacrylamide gel electrophoresis. Protein concentration was determined via the Bradford protein assay (BIO-RAD, Hercules, CA, USA) using bovine serum albumin (BSA) as a standard, and verified via Coomassie Brilliant Blue gel staining. After separating cell lysates (40 μg) via SDS-PAGE (BIO-RAD), protein bands were electro-transferred onto nitrocellulose membranes. Membranes were blocked for 1 h with 5% skimmed milk in Tris-buffered saline (0.137 M NaCl, 0.025 M Tris, pH 7.4) containing 0.1% Tween 20. The following primary antibodies were used in this study: anti-IRE-1α (#3294, Cell Signaling Technology, Danvers, MA, USA), anti-CHOP (#2895, Cell Signaling Technology), anti-β-actin (sc-130300, Santa Cruz Biotechnology, Inc., Santa Cruz, CA, USA), anti-p-PERK (sc-32577, Abcam, Cambridge, MA, USA), anti-PERK (sc-377400, Abcam), anti-p-eIF2α (sc-3398, Abcam), anti-eIF2α (sc-133132, Abcam), anti-SREBP-1 (sc-365513, Abcam), and anti-FAS (sc-8009, Abcam). Horseradish peroxidase (HRP)-conjugated secondary antibodies were obtained from Enzo Life Sciences (Farmingdale, NY, USA). After incubation with an enhanced chemiluminescence (ECL) reagent (BIO-RAD, Hercules, CA, USA), the membranes were exposed to film (Amersham Hyperfilm TM ECL, GE Healthcare Limited, Buckinghamshire, UK) to detect protein signals.

### 2.13. ER Fractionation

The microsomal fraction was obtained as described previously [24]. Briefly, liver tissue was resuspended in buffer A (250 mM sucrose, 20 mM HEPES, pH 7.5, 10 mM KCl, 1.5 mM MgCl_2_, 1 mM EDTA, 1 mM EGTA) and protease inhibitor complex (Roche Diagnostics, Mannheim, Germany) on ice for 30 min. Liver tissue lysates were then homogenized and centrifuged at 750× *g* for 10 min at 4 °C. The supernatant from the homogenized lysates was then centrifuged at 100,000× *g* for 1 h at 4 °C. The resulting supernatant was discarded, and the pellet was stored at −75 °C.

### 2.14. ROS Measurement

Dihydroethidium (DHE) (Life technologies, Grand Island, NY, USA) was used to measure ROS in primary hepatocytes. Cells were treated with 10 μM DHE at 37 °C for 30 min in the dark and images were captured immediately with a fluorescence microscope (Olympus FluoView1000, B&B Microscope Ltd., PA, 40×).

Dihydroethidium (DHE) fluorescence microscopy was performed to detect ROS on frozen liver sections. Cryostat liver sections were cut in 5 μM and incubated with 5 μM DHE for 10 min at room temperature. Images were captured immediately with the same fluorescence microscope (40×).

### 2.15. Statistical Analysis

Study variables are presented as mean ± SEM values. Microcal Origin software (Northampton, MA, USA) was used for all statistical analyses. Student’s *t*-test and one-way ANOVA, followed by a Dunnett’s post hoc test were performed for multiple-group comparisons, with *p* < 0.05 indicating statistical significance.

## 3. Results

### 3.1. Analysis of Compounds in ILF-RE

ILF-R and ILF-E were separately extracted with distilled water. For component analysis, each extract was prepared separately using 70% ILF-R and 50% methanol-ILF-E. In the HPLC analysis of ILF-R (Appendix A), fustin, fisetin, and sulfuretin were identified as the components, with retention times of 9.564, 20.566, and 24.371 min, respectively, and levels of 53.76, 9.34, and 1.94 mg/g, respectively (Appendix A). In the HPLC analysis of ILF-E (Appendix A), geniposide and geniposidic acid were identified as the components, with retention times of 6.118 and 15.969 min, respectively, and levels of 25.97 and 61.91 mg/g, respectively (Appendix A).

### 3.2. ILF-RE Protects against CCl_4_-Induced Liver Damage

To confirm the role of the recently reported combination of herbal products, *R. verniciflua and E. ulmoides* (1:1) [21] in hepatic dysfunction, a rat model of CCl_4_-induced hepatotoxicity was established herein. First, the toxic effects of CCl_4_ were analyzed at different time points. Six hour-treatment of CCl_4_ optimally induced hepatotoxicity in the in vitro model (Appendix A). Treatment with 25, 50, or 100 μg/mL ILF-RE had no effect on hepatocyte viability (Appendix A). After CCl_4_ treatment, hepatocyte viability decreased significantly; however, co-treatment with ILF-RE inhibited cell death in a concentration-dependent manner (Appendix A). To examine the efficacy of the extract, ILF-RE 25, 50, and 100 μg/mL and ILF-RE 100 μg/mL were co-administered during CCl_4_-induced hepatic dysfunction. Combinatorially treated, ILF-RE at 100 μg/mL significantly decreased AST and ALT activity in the hepatocytes compared with only CCl_4_-treated hepatocytes (Figure 1A,B). Since herbal products contain abundant polyphenols and are expected to have strong antioxidant activity [25,26], the free radical scavenging capacity of ILF-RE was examined via the DPPH assay. As shown in Figure 2A, DPPH levels were significantly lower in the CCl_4_-treated group than in the control group. In contrast, DPPH levels increased significantly in primary hepatocytes treated with ILF-RE. In dihydroethidium (DHE) fluorescence-loading cells, CCl_4_ treatment significantly increased ROS production, and ILF-RE treatment markedly attenuated CCl_4_-mediated the increased ROS (Figure 2B), suggesting that ILF-RE protects hepatocytes against CCl_4_-induced damage by attenuating oxidative stress.

### 3.3. ILF-RE Enhances Hepatic Antioxidation in the In Vivo Hepatotoxicity Model

In many biological models, glutathione (GSH) levels are reportedly decreased during oxidative stress [27]. In the present study, the effects of ILF-RE on hepatic GSH levels and antioxidant status were examined. GSH levels were higher in CCl_4_-administered rats treated with ILF-RE than in the rats administered with only CCl_4_ (Figure 3A). The activity of glutathione peroxidase (GPx), superoxide dismutase (SOD), and catalase (CAT) were decreased in the CCl_4_-administered rats, which was recovered in the ILF-RE co-administered group (Figure 3B–D). These results suggest that ILF-RE can protect against CCl_4_-induced liver damage by attenuating oxidative stress.

### 3.4. ILF-RE Protects against Chronic CCl_4_-Induced Hepatic Dysfunction

To determine the effects of ILF-RE on liver injury, we administered 40, 80, or 120 mg/kg of ILF-RE to CCl_4_- or olive oil-administered rats (*n* = 10 rats per group). For the chronic hepatotoxicity model, ILF-RE was administered at 0.2 mL/100 g CCl_4_ twice a week for 4 weeks. Rats with CCl_4_-induced chronic hepatotoxicity exhibited an increase in body weight compared with rats in the control group, and ILF-RE administration in the chronic CCl_4_-induced hepatotoxicity model significantly inhibited weight gain (Appendix A). No significant differences were observed among all groups regarding daily food consumption and fast blood glucose levels (Appendix A). Serum ALT, AST, and GGT levels were significantly elevated in the CCl_4_-induced hepatotoxicity model, but correspondingly decreased upon oral administration of ILF-RE (Figure 4A–C). Liver morphology was also examined via histological analyses. As shown in Figure 4D, diffuse lipid droplets were observed in the livers of mice administered with CCl_4_ but significantly decreased in the CCl_4_–ILF-RE co-administered rats, suggesting that CCl_4_-induced oxidative stress contributes to liver injury and stimulates hepatic steatosis. The effects of ILF-RE on CCl_4_-induced oxidative stress in the liver are shown in Figure 5A,B. In these experiments, red fluorescence from DHE indicates an increase in ROS content in the liver. Chronic CCl_4_ exposure resulted in increased fluorescence, whereas markedly lower fluorescence was observed in the livers of rats treated with ILF-RE. Further, we measured hepatic lipid peroxide levels. The hepatic 4-hydroxynonenal (4-HNE) and lipid peroxide levels were significantly higher in the CCl_4_ group than in the control group (Figure 5C–E). Expectedly, ILF-RE suppressed the increase in 4-HNE and lipid peroxide levels. Similarly, ER-membrane lipid peroxidation was markedly higher in the liver tissue of the CCl_4_-treated rats than in the control rats. As expected, ILF-RE in turn attenuated an increase in the levels of these lipid peroxides (Figure 5F), indicating that ILF-RE inhibits hepatic ROS accumulation in CCl_4_-induced hepatotoxicity.

### 3.5. ILF-RE Regulates Hepatic Lipid Accumulation during ER Stress and Associated Lipogenesis 

In the SREBP-1 lipogenic pathway, alteration of ER folding capacitance and its associated ER stress signaling factors IRE1α, PERK, and ATF6 are interlinked in hepatic dyslipidemia [28,29], and ILF-RE was administered to a model of CCl_4_-induced ER stress and its associated lipogenesis. *E. ulmoides* extracts (EUE) reportedly exert anti-hyperlipidemic and anti-ER stress effects [30]; however, the mechanisms underlying these effects upon combinatorial administration of ILF-R and ILF-E are unknown. To investigate the ability of ILF-RE to regulate hepatic lipid accumulation, a hepatic lipid accumulation assay was performed. Treatment with ILF-RE significantly inhibited CCl_4_-induced cellular lipid accumulation (Figure 6A). Intracellular triglyceride and cholesterol levels were also significantly increased in CCl_4_-treated cells; however, this elevation was attenuated upon ILF-RE treatment (Figure 6B). Among the ER stress signaling proteins, CHOP, P-PERK, and p-eIF2α were significantly upregulated in CCl_4_-treated hepatocytes. Pretreatment with ILF-RE attenuated the upregulation in CHOP, P-PERK, and p-eIF2α (Figure 7A). To investigate the mechanism underlying ILF-RE-mediated reduction in hepatic lipogenesis, we analyzed the expression levels of genes involved in lipid metabolism. During CCl_4_-enhanced hepatic lipogenesis, protein expression of transcription factor SREBP-1 and its target fatty acid synthase (FAS) were significantly attenuated by ILF-RE (Figure 7B). Concurrently, CCl_4_ markedly increased the hepatic lipid content, evident through elevations in triglycerides, total-cholesterol, and LDL-cholesterol, whereas the group co-treated with ILF-RE and CCl_4_ for 4 weeks displayed a significant reduction in triglycerides, total-cholesterol, and LDL-cholesterol (Figure 8A–C). Thus, serum triglyceride, total-cholesterol, and LDL-cholesterol levels were significantly lower in rats administered ILF-RE than in those administered only CCl_4_ (Figure 8D–F). Consistently, lipid droplet accumulation increased significantly in CCl_4_-treated hepatocytes, as determined via Oil Red O staining, although it was markedly reduced upon ILF-RE co-treatment (Figure 8G). In CCl_4_-administered rats, the lipogenic transcription factor SREBP-1 was significantly upregulated, accompanied with upregulation of target lipogenic enzyme FAS (Figure 9A). Furthermore, CHOP, p-PERK, and p-eIF2α were significantly upregulated upon CCl_4_ administration (Figure 9B). Treatment with ILF-RE inhibited the SREBP-1 signaling axis during ER stress, thus elucidating the effect of CCl_4_ administration on the expression of SREBP-1 and CHOP, p-PERK, and p-eIF2α during ER stress signaling. Together, these results indicate that ILF-RE may inhibit lipogenesis and probably improves the ER folding status to ameliorate hepatic lipid accumulation upon CCl_4_ treatment.

## 4. Discussion

The present study shows that chronic CCl_4_-induced liver injury, defined as an increase in the levels of serum markers of hepatic damage and abnormal hepatic lipid accumulation, was attenuated in the presence of ILF-RE. During protection against CCl_4_-mediated hepatotoxicity, antioxidants including GSH, SOD, and catalase were suggested to be upregulated. Regulation of ER stress and its associated lipogenic transcription factor SREBP-1 and downstream target gene FAS are involved in the mechanism underlying ILF-RE-induced protection, explaining how ILF-RE exerts potential protective effects against hepatotoxicity, including hepatic dyslipidemia.

As the main premise of this study, the antioxidant effect of ILF-RE strongly explains the protective effect against CCl_4_-induced hepatic damage. Increased serum ALT, AST, and GGT levels in CCl_4_-treated rats reflect liver damage, as these enzymes leak out from liver into the blood, owing to hepatic tissue damage [31]. Upon ILF-RE administration, the levels of these enzymes were restored, indicating protection against CCl_4_-induced liver damage (Figure 1 and Figure 4). In liver injury, CCl_4_ is biotransformed by the catalytic activity of the liver cytochrome P450 in the ER to generate free radicals, primarily trichloromethyl free radicals (^•^CCl_3_ or CCl_3_OO^•^) as the hepatotoxic metabolites of CCl_4_. These have the potential to bind to various proteins or lipids and initiate lipid peroxidation [32,33], resulting in a loss of cell membrane integrity and liver injury [32], whereas ILF-RE significantly increased the activity of antioxidant enzymes GPx, SOD, and catalase against the hepatotoxic metabolite-associated ROS and damage, thus decreasing serum AST and ALT levels (Figure 3). Since these enzymes serve as the first line of defense to counteract free radical-induced oxidative stress [34], endogenous anti-oxidative enzyme activity is strongly considered one of the primary mechanisms underlying ILF-RE-induced protective effects. Consistently, ILF-E reportedly displays antioxidant activity including that of SOD [30,35], thus also accounting for the hepatoprotective effect against a high-fat diet. Concurrently, other studies have also reported the antioxidant characteristics of EUE [36,37,38]. Furthermore, treatment with only ILF-R reportedly displayed antioxidant activity, and in combination with EUE it exerted synergistic effects of SOD, catalase, and GPx, along with the ratio of GSH/GSSG [21]. This suggests the potential role of antioxidants, while accounting for the protective effect of ILF-RE against hepatotoxicity. 

ILF-RE regulates ER stress and the downstream lipogenic pathway, thus controlling hepatic toxicity and dyslipidemia. In the present study, ROS production and amplification are directly or indirectly associated with the ER, which contains the biotransformation enzyme, CYP2E1. When ROS stimulates an ER redox imbalance and redox-coupling protein, folding is disturbed and the ER elicits an elaborative adaptive response under pathological and/or stressful conditions, collectively known as the UPR [39,40]. Among the three identified UPR signaling pathways, IRE1α reportedly contributes to the development of hepatic steatosis by upregulating genes involved in fatty acid and triglyceride synthesis by XBP-1 [41]. PERK also upregulates lipogenic enzymes such as FAS, ATP citrate lyase, and stearoyl-CoA desaturase-1 in the animal model [42]. Furthermore, CHOP deficiency in mice inhibits cholestasis-induced liver fibrosis [43]. The present results indicate that ILF-RE significantly downregulated CHOP, p-PERK, and p-eIF2, suggesting that ILF-RE potentially protected against hepatic steatosis by regulating the branches of the UPR (Figure 7A and Figure 9A). Furthermore, SREBP and its downstream lipogenic genes were regulated upon ILF-RE administration, a consequence of the regulation of ER stress. During ER stress, the UPR axis (and its target transcription factor SREBP) is one of the established mechanisms of action [44]. Upregulation of lipogenic genes in CCl_4_-treated hepatocytes and in the liver of CCl_4-_administered rats were attenuated by ILF-RE, suggesting that ILF-RE potentially ameliorates CCl_4_-induced hepatic steatosis by inhibiting lipogenesis (Figure 7B and Figure 9B). SREBP-1 is the most important transcription factor regulating FAS. Increasing evidence indicates that ER stress plays an important role in hepatic SREBP-1c activation and in the pathogenesis of NAFLD [45]. A recent study reported that ER stress induced hepatic SREBP-1c cleavage in obese ob/ob mice [46]. This study clearly shows that treatment with ILF-RE inhibits the lipogenic axis and SREBP1-lipogenic gene FAS, elucidating the regulatory effects against lipid accumulation.

In conclusion, ILF-RE efficiently regulated ROS, attenuated ER stress and ER stress-associated lipogenic gene expression, and finally regulated hepatic damage and hepatic lipid dyslipidemia. The present experimental evidence supports the use of ILF-RE as a potential therapeutic agent to prevent/treat chronic hepatic dysfunction.

## Figures and Tables

**Figure 1 nutrients-11-00382-f001:**
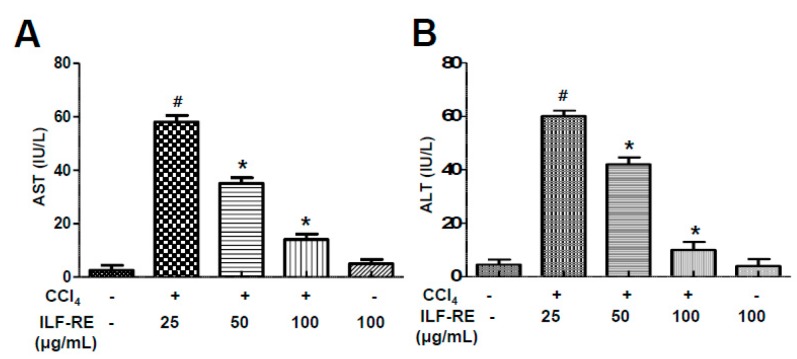
ILF-RE protects against CCl_4_-induced hepatic damage. Primary hepatocytes were treated with 25, 50, and 100 µg/mL of ILF-RE dissolved in water and incubated with 20 mM CCl_4_ for 6 h. Aspartate transaminase (AST) (**A**) and alanine transaminase (ALT) (**B**) activity were determined in hepatocyte primary cells. Data are expressed relative to the controls (*n* = 3; ^#^
*p* < 0.05 vs. Con; * *p* < 0.05 vs. CCl_4_). Con, control; CCl_4_, carbon tetrachloride; ILF-RE, the combination *R. verniciflua* with *E. ulmoides*.

**Figure 2 nutrients-11-00382-f002:**
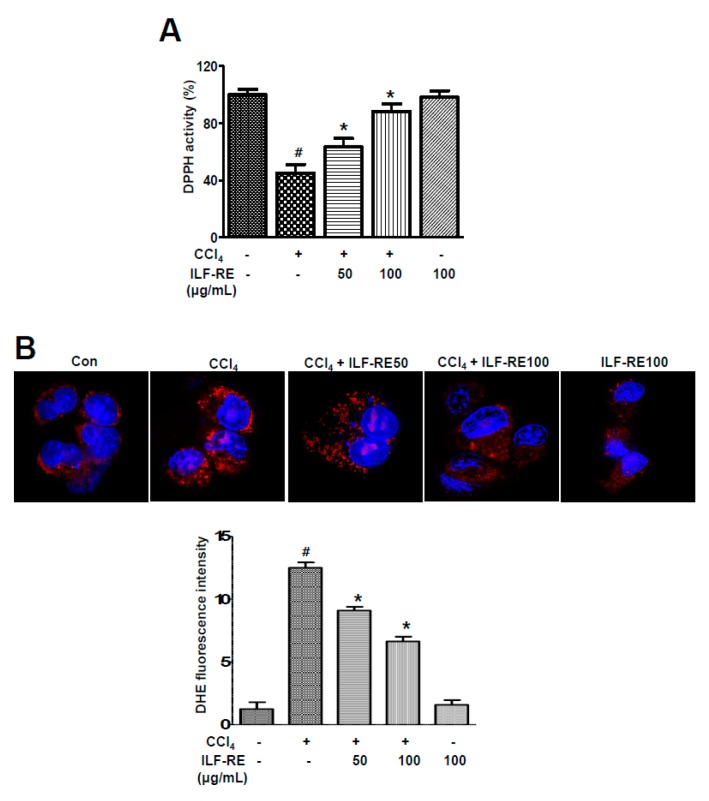
ILF-RE displayed DPPH radical-scavenging activity. Primary hepatocytes were treated with 20 mM CCl_4_ in the absence or presence of 50 and 100 μg/mL ILF-RE for 6 h. (**A**) DPPH radical scavenging was determined in primary hepatocytes. (**B**) Cells were harvested and incubated with 10 μM of DHE at 37 °C for 30 min, washed with PBS, and assessed using a fluorescence microscope. The scale bar represents 50 μM. Data are expressed relative to the controls (*n* = 3; ^#^
*p* < 0.05 vs. Con; * *p* < 0.05 vs. CCl_4_). Con, control; CCl_4_, carbon tetrachloride; ILF-RE, the combination *R. verniciflua* with *E. ulmoides.*

**Figure 3 nutrients-11-00382-f003:**
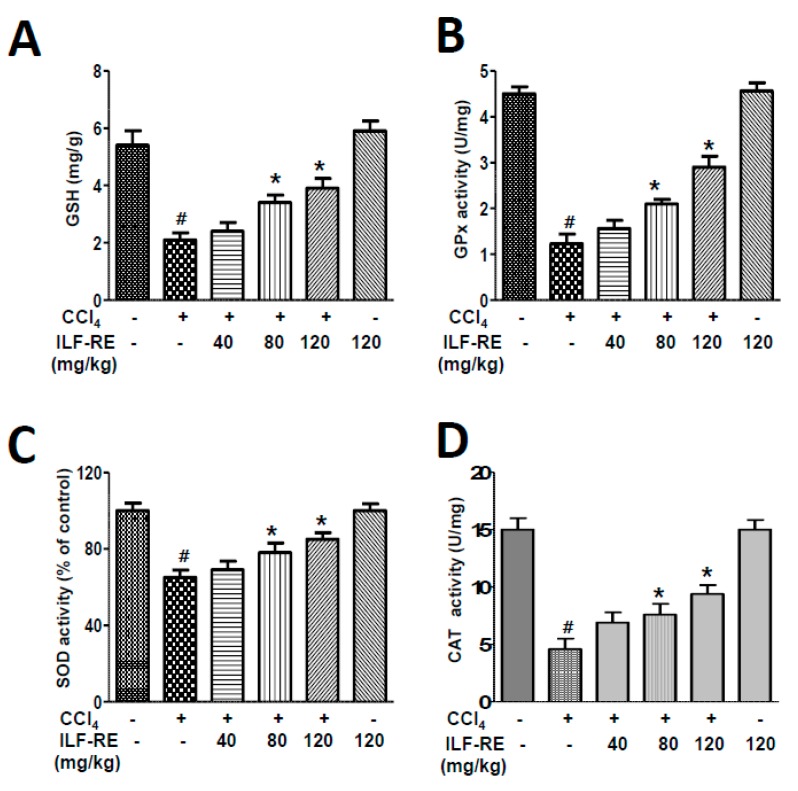
ILF-RE regulates antioxidant activity in CCl_4_-induced hepatotoxicity. Rats were intraperitoneally administered CCl_4_ (0.2 mL/100 g, BW) twice a day for 4 weeks. ILF-R (40, 80, or 120 mg/kg) was co-administered with CCl_4_ or 120 mg/kg ILF-RE only for 4 weeks, followed by harvesting of the livers. GSH (**A**), GPx (**B**), SOD (**C**), and CAT (**D**) activities were determined. The experiments were performed in triplicate with tissues from at least three rats in each group. ^#^
*p* < 0.05 vs. Con group; * *p* < 0.05 vs. CCl_4_ group (*n* = 10 rats per group). BW, body weight; Con, control; CCl_4_, carbon tetrachloride; ILF-RE, the combination *R. verniciflua* (ILF-R) with *E. ulmoides* (ILF-E); GSH, glutathione; GPx, glutathione peroxidase; SOD, superoxide dismutase; CAT: Catalase.

**Figure 4 nutrients-11-00382-f004:**
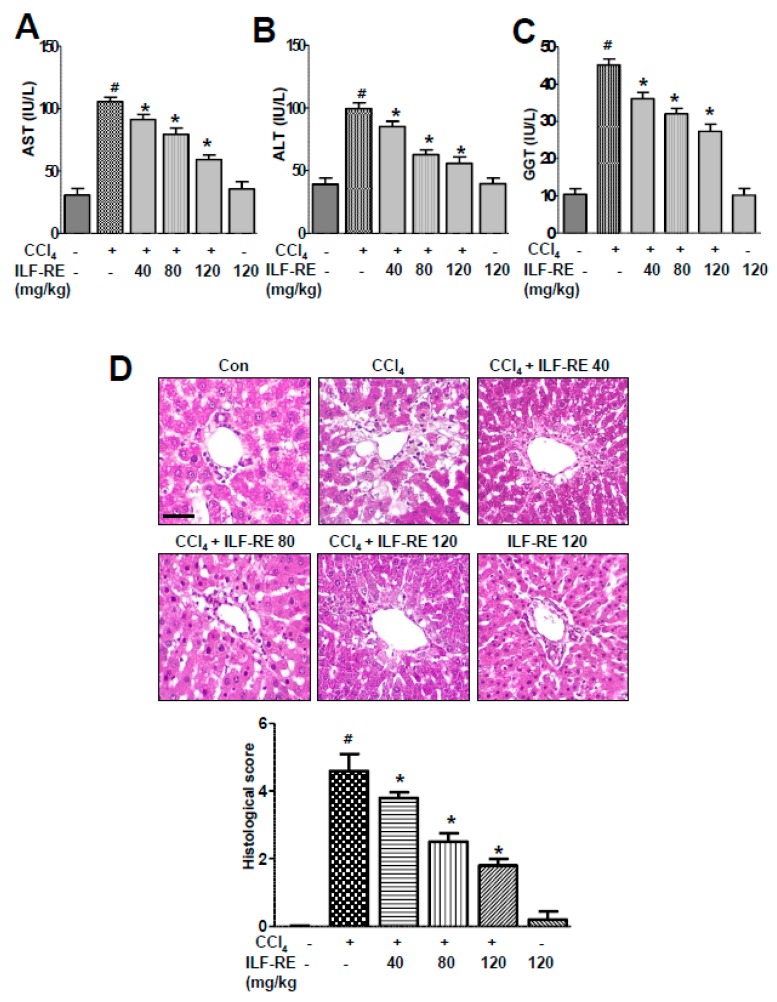
ILF-RE ameliorated CCl_4_-induced toxicity. Rats were intraperitoneally administered CCl_4_ (0.2 mL/100 g, BW) twice a day for 4 weeks. ILF-RE (40, 80, or 120 mg/kg) was co-administered with CCl_4_ or 120 mg/kg ILF-RE only for 4 weeks, followed by harvesting of the livers. Serum AST (**A**), ALT (**B**), and GGT (**C**) levels were determined. (**D**) Representative images of the liver sections from each group, stained with hematoxylin and eosin. Scale bars = 50 µM. The experiments were performed in triplicate with tissues from at least three rats in each group. ^#^
*p* < 0.05 vs. Con group; * *p* < 0.05 vs. CCl_4_ group (*n* = 10 rats per group). Con, control; CCl_4_, carbon tetrachloride; ILF-RE, the combination *R. verniciflua* with *E. ulmoides*; AST, aspartate transaminase; ALT, alanine transaminase; GGT, gamma-glutamyl transpeptidase.

**Figure 5 nutrients-11-00382-f005:**
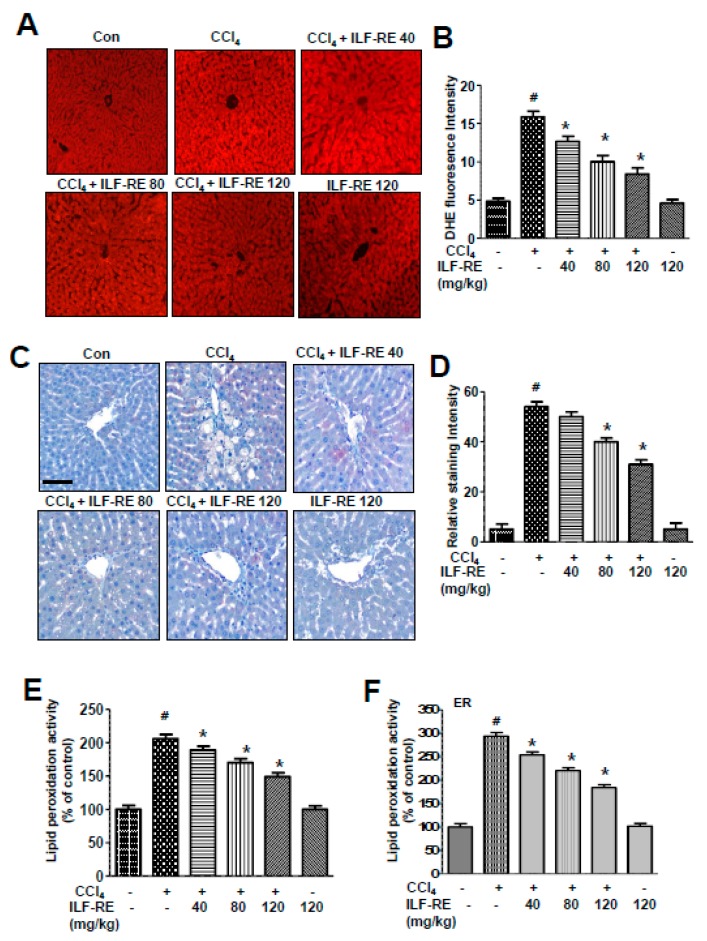
ILF-RE regulates ROS accumulation in CCl_4_-induced hepatotoxicity. Rats were intraperitoneally administered CCl_4_ (0.2 mL/100 g, BW) twice a day for 4 weeks. ILF-RE (40, 80, or 120 mg/kg) was co-administered with CCl_4_ or 120 mg/kg ILF-RE only for 4 weeks, followed by harvesting of the livers. DHE staining (**A**) and quantification assays (**B**) were performed using the livers from these rats. 4-HNE staining was performed (**C**), and the staining intensity of 4-HNE-positive cells was determined (**D**). Lipid peroxidation was assayed in the livers of these rats (**E**) and in the ER fractions from the liver tissues (**F**). The experiments were performed in triplicate using tissues from at least three rats in each group. ^#^
*p* < 0.05 vs. Con group; * *p*  <  0.05 vs. CCl_4_ group (*n* = 10 rats per group). Con, control; CCl_4_, carbon tetrachloride; ILF-RE, the combination *R. verniciflua* with *E. ulmoides*; DHE, dihydroethidium; 4-HNE, 4-hydroxynonenal; ER, endoplasmic reticulum.

**Figure 6 nutrients-11-00382-f006:**
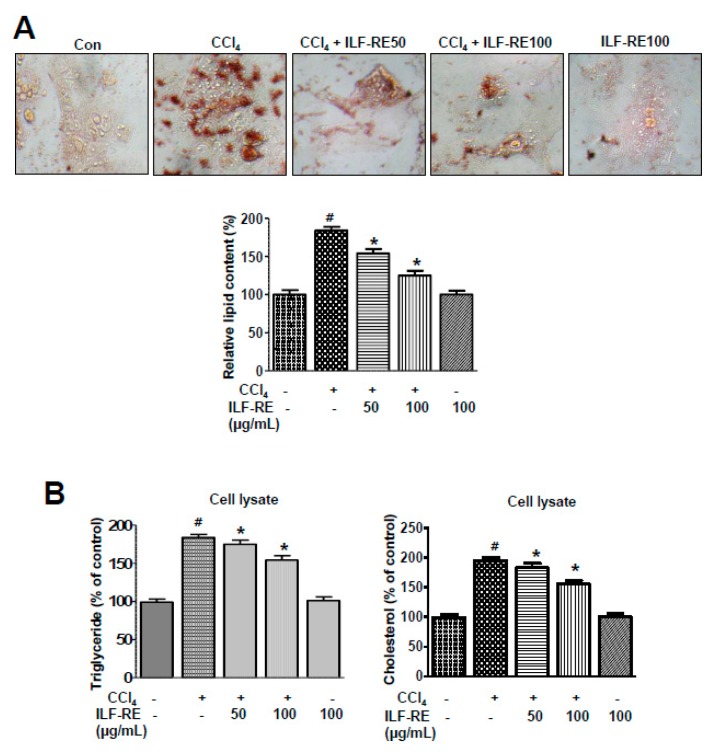
ILF-RE reduced hepatic lipid accumulation. (**A**) Primary hepatocytes were treated with 20 mM CCl_4_ in the absence or presence of 50 and 100 μg/mL ILF-RE for 6 h. Lipid accumulation was assessed via Oil Red O staining. Images of cells were obtained at 200× magnification and used for quantitative analysis of cellular lipid deposition. (**B**) Cells were treated with 50 and 100 μg/mL ILF-RE in the absence or presence of 50 and 100 μg/mL ILF-RE for 6 h. Triglyceride and cholesterol levels were measured in cell lysates. Data are expressed relative to the controls (*n* = 3; ^#^
*p* < 0.05 vs. Con; * *p* < 0.05 vs. CCl_4_). Con, control; CCl_4_, carbon tetrachloride; ILF-RE, the combination *R. verniciflua* with *E. ulmoides.*

**Figure 7 nutrients-11-00382-f007:**
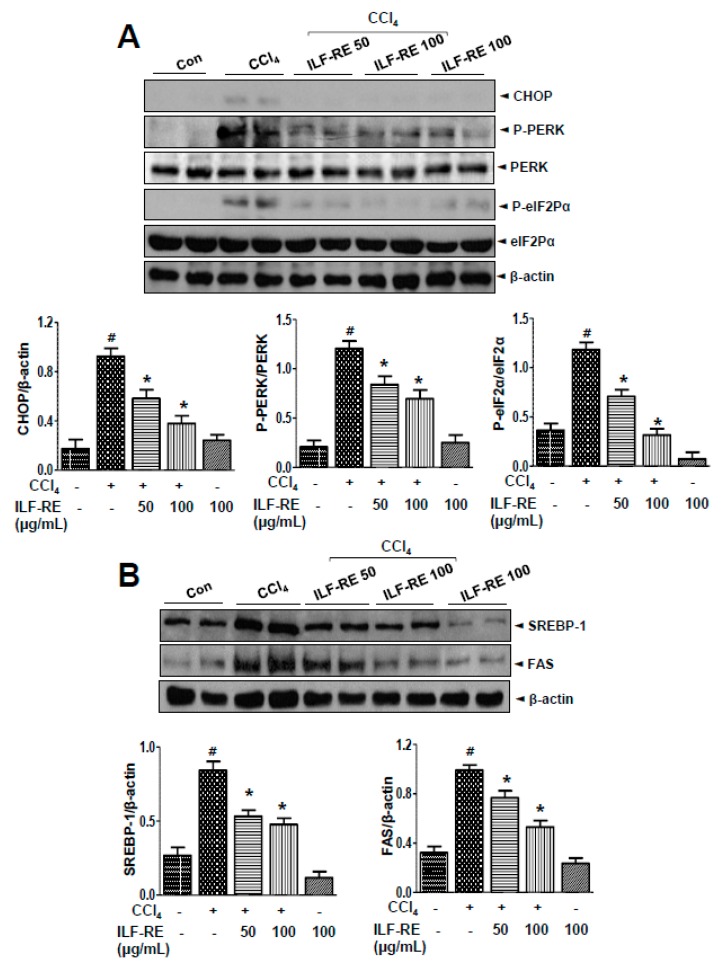
ILF-RE inhibits CCl_4_-induced ER stress and lipogenesis. Primary hepatocytes were treated with 20 mM CCl_4_ in the absence or presence of 50 and 100 μg/mL ILF-RE for 6 h. (**A**) Immunoblotting was performed using anti-CHOP, anti-p-PERK, anti-PERK, anti-p-eIF2α, anti-eIF2α, or anti-β-actin antibodies. Quantification of immunoblot data was performed (lower panel). (**B**) Immunoblotting was performed using anti-SREBP-1, anti-FAS, and anti-β-actin antibodies. Quantification of immunoblot data was performed (lower panel). Data are expressed relative to the controls (*n* = 3; ^#^
*p* < 0.05 vs. Con; * *p* < 0.05 vs. CCl_4_). Con, control; CCl_4_, carbon tetrachloride; ILF-RE, the combination *R. verniciflua* with *E. ulmoides*; ER, endoplasmic reticulum.

**Figure 8 nutrients-11-00382-f008:**
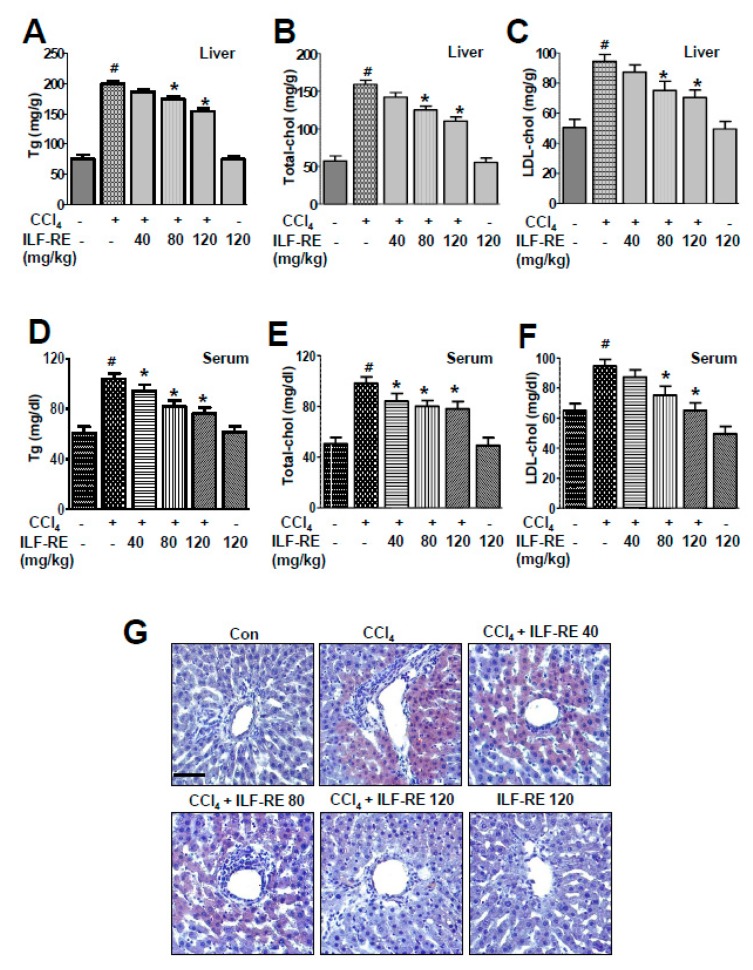
ILF-RE regulates hepatic lipid accumulation in CCl_4_-induced hepatotoxicity. Rats were intraperitoneally treated with CCl_4_ (0.2 mL/100 g, BW) twice a day for 4 weeks. ILF-RE (40, 80, or 120 mg/kg) was co-administered with CCl_4_ or 120 mg/kg ILF-RE on its own for 4 weeks, followed by harvesting of the liver and serum. Liver triglycerides (**A**), liver total-cholesterol (**B**), liver LDL-cholesterol (**C**), serum triglyceride (**D**), serum total-cholesterol (**E**), and serum LDL-cholesterol (**F**) were examined. Representative images of the liver sections from each group stained with Oil Red O for lipid content (**G**). Scale bars = 50 µM. The experiments were performed in triplicate using tissues from at least three rats in each group. ^#^
*p* < 0.05 vs. Con group; * *p* < 0.05 vs. CCl_4_ group (*n* = 10 rats per group). BW, body weight; Con, control; CCl_4_, carbon tetrachloride; ILF-RE, the combination *R. verniciflua* with *E. ulmoides.*

**Figure 9 nutrients-11-00382-f009:**
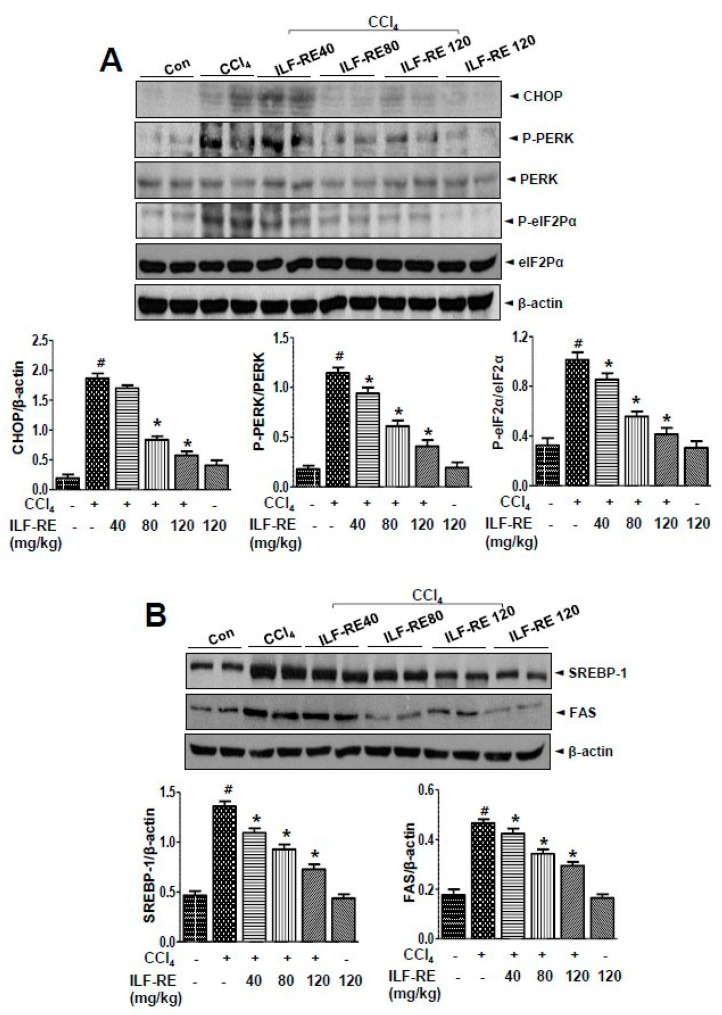
ILF-RE regulated ER stress and lipogenic effect in CCl_4_-induced hepatotoxicity. Rats were intraperitoneally administered CCl_4_ (0.2 mL/100 g, BW) twice a day for 4 weeks. ILF-RE (40, 80, or 120 mg/kg) was co-administered with CCl_4_ or 120 mg/kg ILF-RE on its own for 4 weeks, followed by harvesting of the liver. (**A**) Immunoblotting was performed using anti-SREBP-1, anti-FAS, and anti-β-actin antibodies. Protein expression levels were determined relative to the loading control. (**B**) Immunoblotting was performed using anti-CHOP, anti-p-PERK, anti-PERK, anti-p-eIF2α, anti-eIF2α, and anti-β-actin antibodies. The experiments were performed in triplicate using tissues from at least three rats in each group. ^#^
*p* < 0.05 vs. Con group; * *p* < 0.05 vs. CCl_4_ group (*n* = 10 rats per group). BW, body weight; Con, control; CCl_4_, carbon tetrachloride; ILF-RE, the combination *R. verniciflua* with *E. ulmoides.*

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
