# Peer review of "R. verniciflua and E. ulmoides Extract (ILF-RE) Protects against Chronic CCl4-Induced Liver Damage by Enhancing Antioxidation"

_nutrients, 2019, doi:10.3390/nu11020382_

Reviewer 1 Report

This is a very interesting work with promising results. I really enjoyed reading it. However, It needs some major revisions in Discussion and minor revisions in the other sections.

1- In the abstract you mentioned that chronic hepatic stress was induced by a single

intraperitoneal injection of CCl4 while in the text you said rats were intraperitoneally (i.p.) injected with a mixture of CCl4  and olive oil  twice a week for 4 weeks." It should be corrected.

2- Lines 116-119 explained three different treatments of ILF-RE. Needs to be corrected.

3- Some sentences or words are in bigger sizes. 

4- In the graphs, ILF-RE group in the absence of CCl4 is significant in almost all the results, is there any explain why there are no asterisks on the bars?

5- In the Discussion, you just repeated results while you need do bring literature and discuss your results. Your manuscript has very strong results so you can write an stronger Discussion.

Thanks

Author Response

Rebuttal Letter

This is a very interesting work with promising results. I really enjoyed reading it. However, It needs some major revisions in Discussion and minor revisions in the other sections.

Q1- In the abstract you mentioned that chronic hepatic stress was induced by a single

intraperitoneal injection of CCl4 while in the text you said rats were intraperitoneally (i.p.) injected with a mixture of CCl4  and olive oil  twice a week for 4 weeks." It should be corrected.

A1. According to the reviewer’s comment, we revised the text in the Abstract as follows: “Chronic hepatic stress was induced via intraperitoneal (i.p.) administration of a mixture of CCl4 (0.2 mL/100 g body weight) and olive oil [1:1(v/v)] twice a week for 4 weeks to rats. ILF-RE was administered orally at 40, 80, or 120 mg/kg to rats for 4 weeks.”

Q2- Lines 116-119 explained three different treatments of ILF-RE. Needs to be corrected.

A2. According to the reviewer’s comment, we revised the three treatments of ILF-RE as follows: “Rats were intraperitoneally (i.p.) administered with a mixture of CCl4 (0.2 mL/100 g body weight) and olive oil [1:1(v/v)] twice a week for 4 weeks. Simultaneously, ILF-RE was orally administered daily at dose of 40, 80, or 120 mg/kg body weight.”

Q3- Some sentences or words are in bigger sizes. 

A3. According to the reviewer’s comment, we rectified the sentences and the words to be of uniform size.

Q4- In the graphs, ILF-RE group in the absence of CCl4 is significant in almost all the results, is there any explain why there are no asterisks on the bars?

A4. The CCl4-treated group was significantly different from the control group; hence, we marked a sharp # on the bars. The ILF-RE (40, 80, and 120 mg/kg)-treated groups were significantly different from the CCL4-treated groups; hence, we marked an asterisk * on the indicated ILF-RE treatment group. As expected, even upon single treatment with the highest dose, 120 mg/kg ILF-RE, normal liver functions including hepatic enzymes and hepatic lipid states remained unaltered. The ILF-RE group was not significantly different from the control group in the absence of CCl4 treatment (Fig. 4 and 8). Therefore, we believe that the present data (no asterisks on the bars) are reasonable.

Q5- In the Discussion, you just repeated results while you need do bring literature and discuss your results. Your manuscript has very strong results so you can write an stronger Discussion.

Thanks 

A5. According to the reviewer’s comment, we revised the Discussion as follows: “The present study shows that chronic CCl4-induced liver injury, defined as an increase in the levels of serum markers of hepatic damage and abnormal hepatic lipid accumulation, was attenuated in the presence of ILF-RE. During protection against CCl4-mediated hepatotoxicity, antioxidants including GSH, SOD, and catalase were suggested to be upregulated. Regulation of ER stress and its associated lipogenic transcription factor SREBP-1 and its downstream target gene FAS are involved in the mechanism underlying ILF-RE-induced protection, explaining how ILF-RE exerts potential protective effects against hepatotoxicity including hepatic dyslipidemia.

As the main premise of this study, the antioxidant effect of ILF-RE strongly explains the protective effect against CCl4-induced hepatic damage. Increased serum ALT, AST, and GGT levels in CCl4-treated rats reflect liver damage, as these enzymes leak out from liver into the blood owing to hepatic tissue damage (Rees & Spector, 1961). Upon ILF-RE administration at 40, 80, and 120 mg/kg, the levels of these enzymes were restored, indicating protection against CCl4-induced liver damage (Fig. 1 and 4). In liver injury, CCl4 is biotransformed by the catalytic activity of the liver cytochrome P450 in the ER to generate free radicals, primarily trichloromethyl free radicals (•CCl3 or CCl3OO•) as the hepatotoxic metabolites of CCl4, which have the potential to bind to various proteins or lipids and initiate the lipid peroxidation (Brattin, Glende & Recknagel, 1985; Shah, D'Souza & Iqbal, 2017), resulting in a loss of cell membrane integrity and liver injury (Brattin, Glende & Recknagel, 1985), whereas ILF-RE significantly increased the activity of antioxidant enzymes GPx, SOD, and catalase against the hepatotoxic metabolite-associated ROS and damage, thus decreasing serum AST and ALT levels (Fig. 3). Since these enzymes serve as the first line of defense to counteract free radical-induced oxidative stress (Ozden, Catalgol, Gezginci-Oktayoglu, Arda-Pirincci, Bolkent & Alpertunga, 2009), endogenous anti-oxidative enzyme activity is strongly considered one of the primary mechanisms underlying ILF-RE-induced protective effects. Consistently, ILF-E reportedly displays antioxidant activity including that of SOD (Lee et al., 2014; Lee et al., 2013), thus also accounting for the hepatoprotective effect against a high-fat diet. Concurrently, other studies have also reported the antioxidant characteristics of EUE (Park et al., 2006; Xu, Tang, Li, Liu, Li & Dai, 2010; Yuan, Hussain, Tan, Liu, Ji & Yin, 2017). Furthermore, treatment with only ILF-R reportedly displayed antioxidant activity and in combination with EUE exerted synergistic effects of SOD, catalase, and GPx along with the ratio of GSH/GSSG (Lee, Yoon & Chae, 2018), suggesting the potential role of antioxidants while accounting for the protective effect of ILF-RE against hepatotoxicity. ILF-RE regulates ER stress and the downstream lipogenic pathway, thus controlling hepatic toxicity and dyslipidemia. In the present study, ROS production and amplification are directly or indirectly associated with the ER, which contains the biotransformation enzyme, CYP2E1. When ROS stimulates an ER redox imbalance and redox-coupling protein folding is disturbed, the ER elicits an elaborative adaptive response under pathological and/or stressful conditions, collectively known as the UPR (Fu, Watkins & Hotamisligil, 2012; Ron & Walter, 2007). Among the three identified UPR signalling pathways, IRE1α reportedly contributes to the development of hepatic steatosis by upregulating genes involved in fatty acid and triglyceride synthesis by XBP-1 (Lee, Scapa, Cohen & Glimcher, 2008). PERK also upregulates lipogenic enzymes such as FAS, ATP citrate lyase, and stearoyl-CoA desaturase-1 in the animal model (Bobrovnikova-Marjon et al., 2008). Furthermore, CHOP deficiency in mice inhibited cholestasis-induced liver fibrosis (Tamaki et al., 2008). The present results indicate that ILF-RE significantly downregulates CHOP, p-PERK, and p-eIF2, suggesting that ILF-RE potentially protects against hepatic steatosis by regulating the branches of the UPR (Fig. 7A and 9A). Furthermore, SREBP and its downstream lipogenic genes were regulated upon ILF-RE administration, a consequence of the regulation of ER stress. During ER stress, the UPR axis and its target transcription factor SREBP is one of the established mechanisms of action (Zhang et al., 2012). Upregulation of lipogenic genes in CCl4-treated hepatocytes and in the liver of CCl4-administered rats was attenuated by ILF-RE, suggesting that ILF-RE potentially ameliorates CCl4-induced hepatic steatosis by inhibiting lipogenesis (Fig. 7B and 9B). SREBP-1 is the most important transcription factor regulating FAS. Increasing evidence indicates that ER stress plays an important role in hepatic SREBP-1c activation and in the pathogenesis of NAFLD (Lee, Mendez, Heng, Yang & Zhang, 2012). A recent study reported that ER stress induced hepatic SREBP-1c cleavage in obese ob/ob mice (Kammoun et al., 2009). This study clearly shows that treatment with ILF-RE inhibited the lipogenic axis; SREBP1-lipogenic gene FAS, elucidating the regulatory effects against lipid accumulation.

In conclusion, ILF-RE efficiently regulated ROS, attenuated ER stress and ER stress-associated lipogenic gene expression, and finally regulated hepatic damage and hepatic lipid dyslipidemia. The present experimental evidence supports the use of ILF-RE as a potential therapeutic agent to prevent/treat chronic hepatic dysfunction.”

Reference

Bobrovnikova-Marjon E, Hatzivassiliou G, Grigoriadou C, Romero M, Cavener DR, Thompson CB, et al. (2008). PERK-dependent regulation of lipogenesis during mouse mammary gland development and adipocyte differentiation. Proceedings of the National Academy of Sciences of the United States of America 105: 16314-16319.

Brattin WJ, Glende EA, Jr., & Recknagel RO (1985). Pathological mechanisms in carbon tetrachloride hepatotoxicity. J Free Radic Biol Med 1: 27-38.

Fu S, Watkins SM, & Hotamisligil GS (2012). The role of endoplasmic reticulum in hepatic lipid homeostasis and stress signaling. Cell metabolism 15: 623-634.

Kammoun HL, Chabanon H, Hainault I, Luquet S, Magnan C, Koike T, et al. (2009). GRP78 expression inhibits insulin and ER stress-induced SREBP-1c activation and reduces hepatic steatosis in mice. The Journal of clinical investigation 119: 1201-1215.

Lee AH, Scapa EF, Cohen DE, & Glimcher LH (2008). Regulation of hepatic lipogenesis by the transcription factor XBP1. Science 320: 1492-1496.

Lee GH, Lee MR, Lee HY, Kim SH, Kim HK, Kim HR, et al. (2014). Eucommia ulmoides cortex, geniposide and aucubin regulate lipotoxicity through the inhibition of lysosomal BAX. PloS one 9: e88017.

Lee H, Yoon Y, & Chae H (2018). The effects of herbal extracts on CCl4-induced ROS accumulation and cell death in hepatocytes. Oriental Pharmacy and Experimental Medicine 18: 257-264.

Lee HY, Lee GH, Lee MR, Kim HK, Kim NY, Kim SH, et al. (2013). Eucommia ulmoides Oliver extract, aucubin, and geniposide enhance lysosomal activity to regulate ER stress and hepatic lipid accumulation. PloS one 8: e81349.

Lee JS, Mendez R, Heng HH, Yang ZQ, & Zhang K (2012). Pharmacological ER stress promotes hepatic lipogenesis and lipid droplet formation. Am J Transl Res 4: 102-113.

Ozden S, Catalgol B, Gezginci-Oktayoglu S, Arda-Pirincci P, Bolkent S, & Alpertunga B (2009). Methiocarb-induced oxidative damage following subacute exposure and the protective effects of vitamin E and taurine in rats. Food and chemical toxicology : an international journal published for the British Industrial Biological Research Association 47: 1676-1684.

Park SA, Choi MS, Jung UJ, Kim MJ, Kim DJ, Park HM, et al. (2006). Eucommia ulmoides Oliver leaf extract increases endogenous antioxidant activity in type 2 diabetic mice. Journal of medicinal food 9: 474-479.

Rees KR, & Spector WG (1961). Reversible nature of liver cell damage due to carbon tetrachloride as demonstrated by the use of phenergan. Nature 190: 821-822.

Ron D, & Walter P (2007). Signal integration in the endoplasmic reticulum unfolded protein response. Nature reviews Molecular cell biology 8: 519-529.

Shah MD, D'Souza UJA, & Iqbal M (2017). The potential protective effect of Commelina nudiflora L. against carbon tetrachloride (CCl4)-induced hepatotoxicity in rats, mediated by suppression of oxidative stress and inflammation. Environ Health Prev Med 22: 66.

Tamaki N, Hatano E, Taura K, Tada M, Kodama Y, Nitta T, et al. (2008). CHOP deficiency attenuates cholestasis-induced liver fibrosis by reduction of hepatocyte injury. Am J Physiol Gastrointest Liver Physiol 294: G498-505.

Xu Z, Tang M, Li Y, Liu F, Li X, & Dai R (2010). Antioxidant properties of Du-zhong (Eucommia ulmoides Oliv.) extracts and their effects on color stability and lipid oxidation of raw pork patties. Journal of agricultural and food chemistry 58: 7289-7296.

Yuan D, Hussain T, Tan B, Liu Y, Ji P, & Yin Y (2017). The Evaluation of Antioxidant and Anti-Inflammatory Effects of Eucommia ulmoides Flavones Using Diquat-Challenged Piglet Models. Oxidative medicine and cellular longevity 2017: 8140962.

Zhang C, Chen X, Zhu RM, Zhang Y, Yu T, Wang H, et al. (2012). Endoplasmic reticulum stress is involved in hepatic SREBP-1c activation and lipid accumulation in fructose-fed mice. Toxicol Lett 212: 229-240.

Reviewer 2 Report

To authors,

In this manuscript, the authors evaluated the effect of Rhus verniciflua and Eucommia ulmoides extracts (ILF-RE) on CCl4-induced liver damage. They tested the effect of ILF-RE in both primary hepatocytes and rats models. They showed that ILF-RE can reduce CCl4-induced cell damage, oxidative stress, and lipid accumulation. Mechanistically, they showed that ILF-RE reduced CCl4-induced ER stress and lipogenesis. The authors provided quite a substantial amount of results, however, in the present form, it is recommended this paper be published elsewhere. The following major problems need to be addressed:

1. The description is inconsistent throughout the manuscript. For example, Line 15-16 described that “chronic hepatic stress was induced by single IP injection of CCl4…”, however, Line 115-116 told totally different procedure. Line 118 mentioned that rats were fed with high-fat diet, however, no sign of high-fat diet been used in all the results in this paper!

2. What is the rational to combine Rhus verniciflua and Eucommia ulmoides for treating liver damage? There are lots of plant extracts have been reported to possess good liver protective effects. So, why combined these two plants? It seems this study is not a hypothesis driven study.

3. Line 83 mentioned that “Extracts were prepared using water extracts of ILF-R and ILF-E”, however, in section 3.1, the extracts used here were prepared from 70% and 50% methanol. And these methanolic extracts was used for HPLC analyses. If water extracts were used in all the following experiments, why HPLC analyses of methanolic extracts were showed in Figure 1? The chemical profiles will be hugely different between water and methanolic extracts.

4. Line 117-118 looks like two repeated sentences. Moreover, it is difficult to tell whether CCl4 and ILF-RE extracts were administrated in the same period or CCl4 was injected 4 weeks earlier than ILF-RE treatment.

5. Once been oxidized, dihydroethidium (DHE) shows red fluorescent upon DNA intercalation. Thus, a nuclear staining pattern (such as Figure 5A) was frequently observed in DHE staining assay. However, Figure 2B showed a puncta pattern in cytoplasm, can authors explain that?

6. Line 274, the reduction in lipid peroxidation dose not necessarily correlate to the inhibition of lipogenesis. And just as authors mentioned above, it could be resulted from the reduction of ROS.

7. Lipid accumulates in the cytoplasm as lipid droplets. The staining pattern of lipid droplet is punctual. Thus, the result showed in Figure 8G is not quite convincing.

8. What is ILF stand for?

Author Response

Rebuttal Letter

In this manuscript, the authors evaluated the effect of Rhus verniciflua and Eucommia ulmoides extracts (ILF-RE) on CCl4-induced liver damage. They tested the effect of ILF-RE in both primary hepatocytes and rats models. They showed that ILF-RE can reduce CCl4-induced cell damage, oxidative stress, and lipid accumulation. Mechanistically, they showed that ILF-RE reduced CCl4-induced ER stress and lipogenesis. The authors provided quite a substantial amount of results, however, in the present form, it is recommended this paper be published elsewhere. The following major problems need to be addressed:

Q1. The description is inconsistent throughout the manuscript. For example, Line 15-16 described that “chronic hepatic stress was induced by single IP injection of CCl4…”, however, Line 115-116 told totally different procedure. Line 118 mentioned tha rats were fed with high-fat diet, however, no sign of high-fat diet been used in all the results in this paper!

A1. Thank you for pointing out this critical point. According to the reviewer’s comment, we made revised the Abstract as follows: “Chronic hepatic stress was induced via intraperitoneal (i.p.) administration of a mixture of CCl4 (0.2 mL/100 g body weight) and olive oil [1:1(v/v)] twice a week for 4 weeks to rats. ILF-RE was administered orally at 40, 80, or 120 mg/kg to rats for 4 weeks.”

In addition, we made revised the Materials and Methods section as follows: “Rats were intraperitoneally (i.p.) administered with a mixture of CCl4 (0.2 mL/100 g body weight) and olive oil [1:1(v/v)] twice a week for 4 weeks. Simultaneously, ILF-RE was orally administered daily at dose of 40, 80, or 120 mg/kg body weight.”

Q2. What is the rational to combine Rhus verniciflua and Eucommia ulmoides for treating liver damage? There are lots of plant extracts have been reported to possess good liver protective effects. So, why combined these two plants? It seems this study is not a hypothesis driven study.

A2. In accordance with your comment, we revised the Introduction as follows: “We previously reported that a combined formulation of RV and EU extracts, named ILF-RE herein, displayed synergistic protective effects as opposed to each individual extract (RV or EU) in an in vitro model of CCl4-induced hepatotoxicity [1]. However, the efficacy of the combined extract still needs to be determined in an in vivo model of hepatotoxicity along with its detailed underlying mechanism.”

Q3. Line 83 mentioned that “Extracts were prepared using water extracts of ILF-R and ILF-E”, however, in section 3.1, the extracts used here were prepared from 70% and 50% methanol. And these methanolic extracts was used for HPLC analyses. If water extracts were used in all the following experiments, why HPLC analyses of methanolic extracts were showed in Figure 1? The chemical profiles will be hugely different between water and methanolic extracts.

A3. We appreciate your technical concern. We rectified the error in subsection 3.1 section and revised it as follows: “ILF-R and ILF-E were separately extracted with distilled water. For component analysis, each extract was prepared separately, using 70% (ILF-R) and 50% methanol (ILF-E).”

Q4. Line 117-118 looks like two repeated sentences. Moreover, it is difficult to tell whether CCl4 and ILF-RE extracts were administrated in the same period or CCl4 was injected 4 weeks earlier than ILF-RE treatment.

A4. In accordance with your comment, we revised the subsection on Animal Treatment and Care as follows: “Rats were intraperitoneally (i.p.) administered with a mixture of CCl4 (0.2 mL/100 g body weight) and olive oil [1:1(v/v)] twice a week for 4 weeks. Simultaneously, ILF-RE was orally administered daily at dose of 40, 80, or 120 mg/kg body weight.”

Q5. Once been oxidized, dihydroethidium (DHE) shows red fluorescent upon DNA intercalation. Thus, a nuclear staining pattern (such as Figure 5A) was frequently observed in DHE staining assay. However, Figure 2B showed a puncta pattern in cytoplasm, can authors explain that?

A5. In accordance with the reviewer’s comment, we carefully checked our method. Dihydroethidium, also known as hydroethidine, is a dye that can permeate viable cells and be accumulated in the nucleus after it is dehydrogenated to ethidium bromide. However, due to its desirable quality of passively diffusing into cells, along with its high reactivity, the DHE has been also used to detect cytosolic superoxide and has be used to measure intracellular H2O2 and O2-, and the other reactive oxygens [2-5].

We explain the ROS measurement process in detail as follows;

In original FIGURE 2B, ROS were measured in primary hepatocytes using DHE (Life technologies, Grand Island, NY, USA). Images were captured using a fluorescence microscope at 40X magnification.

However, in original Figure 5A, DHE fluorescence in frozen liver section was measured using a fluorescence microscope at 200X magnification. Because the image was from a high magnification (200X), it looks as if only the nucleus was stained.

In the revised Figure 5A, the DHE stained liver section was photographed under a low-magnification (X40) instead of the high magnification X200.

In addition, we added the missing information of DHE staining analysis in the revised MATERIALS AND METHODS section as follows:

2.14. ROS measurement

Dihydroethidium (DHE) (Life technologies, Grand Island, NY, USA) was used to measure ROS in primary hepatocytes. Cells were treated with 10 μM DHE at 37 °C for 30 min in dark. Images were captured immediately with a fluorescence microscope (Olympus FluoView1000, B&B Microscope Ltd., PA, 40X).

Dihydroethidium (DHE) fluorescence microscopy was performed to detect ROS on frozen liver sections. Cryostat liver sections were cut in 5 μm and incubated with 5 μM DHE for 10 min at room temperature. Images were captured immediately with the same fluorescence microscope (40X).

And we updated FIURE 5A as follows;

Q6. Line 274, the reduction in lipid peroxidation dose not necessarily correlate to the inhibition of lipogenesis. And just as authors mentioned above, it could be resulted from the reduction of ROS.

A6. A6. In accordance with your comment, we revised this description as follows: “As expected, ILF-RE in turn attenuated the increase in the levels of these lipid peroxides (Fig. 5F), indicating that ILF-RE inhibits hepatic ROS accumulation in CCl4-induced hepatotoxicity.”

Q7. Lipid accumulates in the cytoplasm as lipid droplets. The staining pattern of lipid droplet is punctual. Thus, the result showed in Figure 8G is not quite convincing.

A7. We appreciate your technical concern. In accordance with your comment, we revised the data regarding hepatic lipid staining as follows:

Q8. What is ILF stand for?

A8. “ILF” stands for “Imsil Liver Function.” An explanation for ILF has been provided in the Acknowledgments section.

Reference

1.       Lee, H.; Yoon, Y.; Chae, H. The effects of herbal extracts on CCl4-induced ROS accumulation and cell death in hepatocytes. Oriental Pharmacy and Experimental Medicine 2018, 18, 257-264, doi:10.1007/s13596-018-0318-x.

2.       Gomes, A.; Fernandes, E.; Lima, J.L. Fluorescence probes used for detection of reactive oxygen species. J Biochem Biophys Methods 2005, 65, 45-80, doi:10.1016/j.jbbm.2005.10.003.

3.       Wojtala, A.; Bonora, M.; Malinska, D.; Pinton, P.; Duszynski, J.; Wieckowski, M.R. Methods to monitor ROS production by fluorescence microscopy and fluorometry. Methods Enzymol 2014, 542, 243-262, doi:10.1016/B978-0-12-416618-9.00013-3.

4.       Wardman, P. Fluorescent and luminescent probes for measurement of oxidative and nitrosative species in cells and tissues: progress, pitfalls, and prospects. Free radical biology & medicine 2007, 43, 995-1022, doi:10.1016/j.freeradbiomed.2007.06.026.

5.       Zhao, H.; Kalivendi, S.; Zhang, H.; Joseph, J.; Nithipatikom, K.; Vasquez-Vivar, J.; Kalyanaraman, B. Superoxide reacts with hydroethidine but forms a fluorescent product that is distinctly different from ethidium: potential implications in intracellular fluorescence detection of superoxide. Free radical biology & medicine 2003, 34, 1359-1368.

Round  2

Reviewer 2 Report

To author

This is a much improved manuscript. Still, one point I would suggest:

The control panel in Figure 5A should be replaced with other image, the morphology of that liver section was just not like "normal". 

Author Response

Jan. 28. 2019

Editor

Nutrients

Dear

Thank you very much for your letter regarding our manuscript: “Rhus verniciflua and Eucommia ulmoides extract (ILF-RE) protects against chronic CCl4-induced liver damage by enhancing antioxidation” in Nutrients. We examined the reviewers’ comments carefully and have made corrections accordingly. We thank you very much editor and reviewers for constructive and valuable comments. I hope this revised manuscript will now be suitable for publication in Nutrients.

Thank you very much for your consideration and we look forward to your reply.

Sincerely yours,

Han-Jung Chae, PhD

Professor

Department of Pharmacology

School of Medicine

Chonbuk National University

San 2-20, Geum-Am Dong, Duk-Jin Ku

Jeonju, 561-181, Republic of Korea

Tel: 82-63-270-3092

Fax: 82-63-275-8799

Rebuttal Letter

Q1. This is a much improved manuscript. Still, one point I would suggest:

The control panel in Figure 5A should be replaced with other image, the morphology of that liver section was just not like "normal". 

A1. In accordance with your comment, we revised the data as follows;
